# Sex-Specific Anxiety and Prefrontal Cortex Glutamatergic Dysregulation Are Long-Term Consequences of Pre-and Postnatal Exposure to Hypercaloric Diet in a Rat Model

**DOI:** 10.3390/nu12061829

**Published:** 2020-06-19

**Authors:** Patricia Rivera, Rubén Tovar, María Teresa Ramírez-López, Juan Antonio Navarro, Antonio Vargas, Juan Suárez, Fernando Rodríguez de Fonseca

**Affiliations:** 1Instituto de Investigación Biomédica de Málaga (IBIMA), Hospital Regional Universitario de Málaga, Universidad de Málaga, 29010 Málaga, Spain; rubentovar7@hotmail.com (R.T.); juan_naga@hotmail.es (J.A.N.); antoniovargasfuentes@gmail.com (A.V.); juan.suarez@ibima.eu (J.S.); 2Hospital Universitario de Getafe, Servicio de Ginecología y Obstetricia, 28905 Getafe, Spain; maytralop@yahoo.es; 3Departamento de Enfermería, Facultad de Enfermería, Fisioterapia y Podología, Universidad Complutense de Madrid, 28040 Madrid, Spain

**Keywords:** perinatal programming, prefrontal cortex, endocannabinoid system

## Abstract

Both maternal and early life malnutrition can cause long-term behavioral changes in the offspring, which depends on the caloric availability and the timing of the exposure. Here we investigated in a rat model whether a high-caloric palatable diet given to the mother and/or to the offspring during the perinatal and/or postnatal period might dysregulate emotional behavior and prefrontal cortex function in the offspring at adult age. To this end, we examined both anxiety responses and the mRNA/protein expression of glutamatergic, GABAergic and endocannabinoid signaling pathways in the prefrontal cortex of adult offspring. Male animals born from mothers fed the palatable diet, and who continued with this diet after weaning, exhibited anxiety associated with an overexpression of the mRNA of *Grin1*, *Gria1* and *Grm5* glutamate receptors in the prefrontal cortex. In addition, these animals had a reduced expression of the endocannabinoid system, the main inhibitory retrograde input to glutamate synapses, reflected in a decrease of the *Cnr1* receptor and the *Nape-pld* enzyme. In conclusion, a hypercaloric maternal diet induces sex-dependent anxiety, associated with alterations in both glutamatergic and cannabinoid signaling in the prefrontal cortex, which are accentuated with the continuation of the palatable diet during the life of the offspring.

## 1. Introduction

Maternal nutrition plays a crucial role in the offspring’s brain development during pregnancy and during a critical period after birth. Initial studies on perinatal programming have shown that malnutrition during pregnancy and/or lactation induces metabolic disorders such as hyperphagia, adiposity, hyperlipidemia and glucose intolerance in offspring, even into adulthood [1,2,3,4]. Moreover, growing evidence has revealed a modulation of the offspring behavior related to depression, anxiety-like behaviors and addiction (preference of hypercaloric foods and drugs) due to the consumption of a hypercaloric diet in the perinatal period, being associated with long-term structural and functional alterations in the neurodevelopment of the prefrontal cortex (PFC) [5,6,7,8].

Behavioral disorders related to chronic stress/insults are associated with changes in glutamatergic and GABAergic neurotransmission in the hippocampus and prefrontal cortex [9,10,11]. The adequate modulation of these neurotransmitters is essential for the adaptive behavioral response. During stressful experiences, including nutritional insult, there is neuroplasticity of glutamatergic signaling in the brain areas involved in stress response. An excessive glutamate activity and/or lack of gamma-aminobutyric acid (GABA) can generate negative associative memories, being associated with depression and anxiety [12,13,14].

A fatty diet can be considered a stressor affecting central neurotransmission, especially in areas directly related to metabolic control, such as the hypothalamus or the hippocampus, but also in areas that control eating behavior, such as the prefrontal cortex [15]. A high-fat diet during the perinatal period affects the cognitive functions of the offspring at different stages of age, and may be attributed to neurochemical changes in the hippocampus and PFC [16].

There are a set of homeostatic processes that control the release of neurotransmitters after neuronal activation; the most highly characterized is the endocannabinoid system (ECS), comprising the endogenous lipids anandamide (AEA, arachidonoylethanolamide) and 2-arachidonoylglycerol (2-AG), their target cannabinoid (CB) receptors CB1 and CB2, and their synthetic and catabolic enzymes. ECS is a retrograde neuromodulatory system of the major excitatory (glutamatergic) and inhibitory (GABAergic) synapses in the brain. Endogenous cannabinoids are released postsynaptically, and bind to presynaptic cannabinoid receptors, thus inhibiting the release of glutamate or GABA [17,18]. It is known that in the face of acute stress, the levels of the endocannabinoid 2-arachidonyl glycerol (2-AG) increase in involved brain areas such as the hippocampus and PFC [19].

ECS develops during prenatal and early postnatal life, in humans and animals, in the PFC. ECS is highly concentrated at interneuron synapses, playing a critical role in the developmental trajectory of GABAergic interneurons [20,21,22]. In addition, several studies have demonstrated that the ECS is altered in different brain areas of the offspring born from mothers with a caloric restriction or a hypercaloric diet during the perinatal period [3,23,24].

Since anxiety disorders have been related to regional dysregulation of excitatory/inhibitory neurotransmission pathways, we investigated the possible implication of glutamatergic and GABAergic signaling in the anxiogenic action of the perinatal and/or postnatal palatable diets, in the PFC of the male and female offspring in adulthood. Given the modulatory role of the ECS in glutamatergic and GABAergic neurotransmission during development, we hypothesized that the expression of the main elements of this homeostatic system are also altered in the adult offspring’s PFC by the perinatal and/or postnatal exposure to a palatable diet.

With this aim, we analyze the expression of the receptors and the enzymes of synthesis and degradation of the endocannabinoids, as well as the expression of the main glutamatergic and GABAergic receptors in the PFC of the offspring in the adult state, after a perinatal and/or postnatal hypercaloric diet.

## 2. Materials and Methods 

### 2.1. Ethics Statement

Animal experimentation was carried out according to the recommendations of the European Communities Directive 2010/63/EU and Spanish legislation (Real Decreto 53/2013, BOE 34/11370-11421, 2013) regulating the care and use of laboratory animals. The protocol was approved by the Animal Ethics Committee of the Complutense University of Madrid (PSI-2012-35388; January 2012).

### 2.2. Animals

Adolescent female Wistar rats (*n* = 20; 191.7 ± 2.6 g; Harlan, Barcelona, Spain) were used, which were housed individually under controlled conditions of light (12-h light–dark cycle), temperature (21 ± 1 °C) and humidity (40% ± 5%). After three weeks of acclimatization, the rats were randomly assigned to a standard chow (*n* = 9) or a free-choice palatable diet (*n* = 11) eight weeks before mating.

### 2.3. Diets

Control rats were given free access to standard chow (SAFE A04, Panlab, Barcelona, Spain). The free-choice diet rats were allowed to choose between standard chow (C) and a free-choice palatable diet (P) containing a mixture of chocolates (cafeteria diet) composed of Mars^®^, Snickers^®^, Bounty^®^ and Milka^®^ in equal proportions [25]. The water and both diets were administered ad lubitum and their composition is shown in Table 1.

### 2.4. Experimental Design

After 8 weeks of pregestation with the assigned diets, the females were mated for 24 h from the start of the proestrus. Gestational day 0 (GD0) was determined by the presence of sperm. The dietary paradigm was maintained during pregnancy and lactation. At 14 h after birth (PND0), pups were weighed and sexed. The litter size was arranged to comprise 8 pups consisting of 4 males and 4 females where possible. The remaining pups were quickly sacrificed by decapitation. Mating was successful in 9 female rats exposed to the C diet, resulting in 9 C litters, and in 11 mothers exposed to the free-choice P diet, resulting in 11 P litters. All litters were equally represented in each offspring group. Initially, we randomly distributed 3–4 pups per litter, resulting in 35–41 pups per experimental group (35 C males, 36 C females, 38 P males and 41 P females). At PND 22–23, all offspring were weaned and randomly assigned to a C diet or free-choice P diet, resulting in eight experimental groups with 1–2 pups per litter (*n* = 15–20): C diet-fed male and female offspring born to C diet-fed dams (15 CC males and 16 CC females); C diet-fed male and female offspring born to free-choice P diet-exposed dams (17 PC males and 21 PC females); free-choice P diet-exposed male and female offspring born to C diet-fed dams (16 CP males and 16 CP females); and free-choice P diet-exposed male and female offspring born to free-choice P diet-exposed dams (17 PP males and 20 PP females). To minimize estrous cycle-related variability, the female offspring were closely housed in adjacent cages and were separated from males [26]. Food intake and body weight were measured daily during pregnancy and lactation and weekly after weaning. At postnatal weeks 21–23, the adult offspring were sacrificed.

### 2.5. Sample Collection

The adult offspring were sacrificed at postnatal weeks 21–23, by decapitation after the administration of Equitesin^®^ (3 mg kg^−1^). Sacrifices were made in an isolated room two hours after the start of the dark phase. Brains were frozen and stored at –80 °C (*n* = 5–6; 1 rat per litter was randomly selected). The prefrontal cortex (PFC) was dissected from Bregma 4.70 mm to 2.20 mm with fine surgical instruments [27]. Brain samples were weighed and stored at −80 °C until they were used for mRNA and protein analysis.

### 2.6. RNA Isolation and Real-Time Quantitative PCR Analysis

Real-time PCR (TaqMan, ThermoFisher Scientific, Waltham, MA, USA) was performed as described previously [28] using TaqMan Gene Expression Assays primers shown in Table 2. Prefrontal cortex RNA (*n* = 5–6; 1 rat per litter was randomly selected) was extracted following the Trizol^®^ method (ThermoFisher Scientific). RNA samples were isolated with the RNeasy MinElute cleanup kit, including digestion with a DNase I column (Qiagen, Hilden, Germany). Reverse transcript reaction of 1 μg of RNA and quantitative real-time reverse transcription polymerase chain reaction (qPCR) were performed in a CFX96TM Real-Time PCR Detection System (Bio-Rad, Hercules, CA, USA) using the FAM dye-labeled format for the TaqMan^®^ Gene Expression Assays (ThermoFisher Scientific). We found that a single product was amplified using a melting curve. The results were normalized in relation to *Actb* after checking their homogeneity between groups.

### 2.7. Western Blot Analysis

Western blotting was performed as described previously [29]. Briefly, prefrontal cortex (*n* = 4; 1 rat per litter were randomly selected) were homogenized in 500 μL of ice-cold lysis buffer containing Triton X-100, 1 M 4-(2-hydroxyethyl)-1-piperazineethanesulfonic acid (HEPES), 0.1 M ethylenediaminetetraacetic acid (EDTA), sodium pyrophosphate, sodium fluoride (NaF), sodium orthovanadate (NaOV) and protease inhibitors using a tissue-lyser system (Qiagen). After centrifuging at 26,000× *g* for 30 min at 4 °C, the supernatant was transferred to a new tube. Bradford method was used to measure the protein concentration of the samples. A quantity of 30 μg of each total protein sample was separated on 4–12% polyacrylamide gradient gels. The gels were then transferred onto nitrocellulose membranes (Bio-Rad Laboratories, Hercules, CA, USA) and stained with Ponceau red, Membranes were blocked in TBS-T (50 mM Tris-HCl pH 7.6, 200 mM NaCl, and 0.1% Tween 20) with 2% albumin fraction V from bovine serum (BSA, Roche, Mannheim, Germany) for 1 h at room temperature. The primary antibodies to the proteins of interest (Table 3) were incubated overnight at 4 °C. Mouse βactin was used as the reference protein. After several washes in TBS containing 1% Tween 20, an HRP-conjugated anti-rabbit or anti-mouse IgG (H+L) secondary antibody (Promega, Madison, WI, USA), diluted 1:10,000, was added followed by incubation for 1 h at room temperature. After extensive washing in TBS-T, the membranes were incubated for 1 min with the Western Blotting Luminol Reagent kit (Santa Cruz Biotechnology, Santa Cruz, CA, USA), and the specific protein bands were visualized and quantified by chemiluminescence using a ChemiDocTM MP Imaging System (Bio-Rad, Barcelona, Spain). The results are expressed as the target protein/ βactin ratios.

### 2.8. Behavioral Studies

At week 7–8 postnatal, anxiety-related behaviors were evaluated in the offspring using the open field and elevated plus maze tests.

The elevated plus-maze (Panlab, Barcelona, Spain) is a cross-shaped platform elevated 65 cm from the floor with two opposing open arms (50 × 10 cm) and two closed arms (50-cm high opaque walls). A central area of 10 cm^2^ connected four arms. The light intensity was set at 150 lux in the open arms and 80 lux in the closed arms. The test was performed at 5 h after the beginning of the dark phase. The animal was allowed to explore the maze freely for 5 min. A computer-controlled system recorded the number of entries and the time spent in each arm. Data were analyzed by using the MAZEsoft software (Panlab, Barcelona, Spain). Animals that fell off the maze during the test were excluded from the analysis.

Two days after the elevated plus maze test, the open field test was carried out, which isa square arena (80 cm × 80 cm and 40 cm high) virtually divided into a peripheral zone and a central zone (40 cm × 40 cm) located in a room with low light intensity (30 lux). The test was performed 5 h after the beginning of the dark phase. Rats explored freely for 5 min. A video tracking motion analysis system (Smart, Panlab, Harvard Apparatus, Spain) measured the total distance traveled (cm) and mean speed (cm/s), and calculated the time spent on central area as well as the number of entries to the central zone as an index of anxiety-like behavior.

### 2.9. Statistical Analysis

All data are expressed as the mean ± SEM. Animal model data were analyzed by three-way ANOVA (sex, maternal diet and offspring diet as factors) using IBM SPSS Statistics 23. Subsequent comparisons between two groups were carried out using Student’s *t*-test. Pearson’s correlation analysis was performed using IBM SPSS Statistics 23. *p* < 0.05 was considered statistically significant.

## 3. Results

### 3.1. Effect of Maternal and/or Postnatal Hypercaloric Diet on Offspring Behavior

The main effects of sex [F (1,106) = 13.842; *p* < 0.001], and maternal [F (1,106) = 5.341; *p* < 0.001] and postnatal diet [F (1,106) = 4.120; *p* < 0.05], and a significant interaction between postnatal diet and sex, were found on the time spent in the central area of the open field test [F (1,29) = 7.347; *p* < 0.05], with the PP male group spending significantly less time in the central area of the arena (* *p* < 0.05; Figure 1A).

There were effects of sex [F (1,120) = 10.465; *p* < 0.01] and postnatal diet [F (1,120) = 6.575; *p* < 0.05] on the time spent in the open arms of the elevated plus maze test, with an interaction of maternal and postnatal diets [F (1,120) = 6.195; *p* < 0.05]. CP females, PC males and PP males and females spent less time in the open arms than CC groups (*^/^** *p* < 0.05/0.01; Figure 1B).

Maternal diet and sex affected the number of entries in the central area of the open field test [F (1,123) = 6.126; *p* < 0.05; F (1,123) = 53.406; *p* < 0.001], with an interaction between maternal and postnatal diets and sex [F (1,123) = 5.946; *p* < 0.05]. PP males and PC females entered the central area of the arena less often (* *p* < 0.05; ^#/##^
*p* < 0.05/0.01; ^&&^
*p* < 0.01; Figure 1C).

The number of entries in the open arm of the elevated plus maze test was affected by sex [F (1,122) = 27.350; *p* < 0.001], with interactions between maternal and postnatal diets [F (1,122) = 7.201; *p* < 0.01] and sex, maternal and postnatal diets [F (1,122) = 4.195; *p* < 0.05]. PC and PP males and CP female groups entered the open arms less often, as compared to control rats (*^/^** *p* < 0.05/0.01; Figure 1D).

Regarding the locomotor activity, a sex effect was found in the total distance travelled in the open field test [F (1,124) = 50.772; *p* < 0.001], with an overall increase in the distance traveled by PP females (* *p* < 0.05; ^#^
*p* < 0.05; ^&^
*p* < 0.05; Figure 1E).

A sex effect was also found in the total entries in the elevated plus maze test [F (1,124) = 23.678; *p* < 0.001], with an overall decrease in the number of entries made by the PP male group (** *p* < 0.01; ^#^
*p* < 0.05; ^&^
*p* < 0.05; Figure 1F).

### 3.2. Effects of Maternal and/or Postnatal Hypercaloric Diet on Glutamatergic Signaling in Adult Offspring Prefrontal Cortex

Regarding glutamate synthesis and transport, an interaction between maternal and postnatal diets was found on the mRNA levels of the glutamate synthesizing enzyme *Gls* [F (1,41) = 8.399; *p* < 0.01], with an overall increase in *Gls* mRNA in PP male animals compared to CP and PC groups (^#^
*p* < 0.05; ^&^
*p* < 0.05; Figure 2A). A significant interaction between postnatal diet and sex was also found on the mRNA levels of glutamate uptake protein *Slc1a1* [F (1,41) = 12.359; *p* < 0.01], with the hypercaloric postnatal diet increasing *Slc1a1* mRNA levels in CP and PP male offspring (*^/^** *p* < 0.05/0.01; ^&^
*p* < 0.05; Figure 2C).

Regarding ionotropic glutamate receptors, a postnatal diet effect and an interaction between postnatal diet and sex were found on the mRNA levels of *Grin1* [F (1,41) = 7.252; *p* < 0.05; F (1,41) = 5.418; *p* < 0.05], with an increase in *Grin1* mRNA levels in PP males compared to the CC group (* *p* < 0.05; Figure 2D). 

A maternal diet effect was found on the mRNA levels of *Grin2c* [F (1,41) = 8.337; *p* < 0.01], with the hypercaloric maternal diet increasing *Grin2c* mRNA levels in PC and PP male offspring (^*/**^
*p* < 0.05/0.01; ^##^
*p* < 0.01; Figure 2G). Main effects of sex [F (1,41) = 4.551; *p* < 0.05], maternal diet [F (1,41) = 7.368; *p* < 0.05] and postnatal diet [F (1,41) = 4.069; *p* < 0.05], and a significant interaction between maternal diet and sex [F (1,41) = 5.236; *p* < 0.05] and postnatal diet and sex [F (1,41) = 5.061; *p* < 0.05] were found on the mRNA levels of *Gria1*, with an increase in *Gria1* mRNA levels found in response to maternal and postnatal hypercaloric diets in the male offspring (PP group) (** *p* < 0.01; ^#^
*p* < 0.05; Figure 2H).

There was an interaction between maternal and postnatal diets on mRNA levels of *Gria2* [F (1,41) = 6.664; *p* < 0.05], with an overall increase in *Gria2* mRNA levels in PP males (* *p* < 0.05; ^##^
*p* < 0.01; ^&^
*p* < 0.05; Figure 2I).

Significant interactions between postnatal diet and sex [F (1,41) = 4.799; *p* < 0.05] and maternal and postnatal diets [F (1,41) = 3.937; *p* < 0.05] were also found on mRNA levels of *Gria4*, with an increase in *Gria4* mRNA levels in PP males compared to CP and PC groups (^###^
*p* < 0.001; ^&^
*p* < 0.05; Figure 2K).

Regarding metabotropic glutamate receptors, there was a significant interaction between postnatal diet and sex on mRNA levels of *Grm5* [F (1,41) = 8.199; *p* < 0.01], with an increase in *Grm5* mRNA levels in CP, PC and PP males. compared to the CC group (^*/**^
*p* < 0.05/0.01; Figure 2M). No main effects or interactions between factors were found in Grm3 expression, however Student’s *t* test showed an increase in *Grm3* mRNA levels in the CP and PP male groups, compared to the control group (* *p* < 0.05; Figure 2L).

In order to establish the relationship between the main changes found in anxiety-related behavioral tests and those found in the mRNA expression of glutamatergic signaling in males, we performed correlation studies.

Our results showed a negative correlation between the time spent in the central area of the open field test and the mRNA expression of *Gls* and *Gria2* (*p* < 0.01).

The time spent in open arms of the elevated plus maze test also correlated negatively with gene expression of the glutamatergic receptors *Gria1* and *Grm5* (*p* < 0.05).

Finally, the number of entries that the animals made into the open arms of the elevated plus maze test showed a negative correlation with the mRNA levels of *Slc1a1* (*p* < 0.05), *Grin1* (*p* < 0.01), Gria1 (*p* < 0.05), Grm3 (*p* < 0.05) and Grm5 (*p* < 0.01). The correlation coefficients are shown in Table 4.

No correlation was found between the number of entries into the central area of the open field test and the glutamatergic system genes included in this part of the study.

### 3.3. Effects of Maternal and/or Postnatal Hypercaloric Diet on Endocannabinoid Signaling in Adult Offspring Prefrontal Cortex

The main effects of maternal [F (1,41) = 6.147; *p* < 0.05] and postnatal diets [F (1,41) = 5.103; *p* < 0.05], and a significant interaction between both diets, were found on the mRNA levels of the cannabinoid receptor *Cnr1* [F (1,41) = 5.779; *p* < 0.05], with a decrease in the Cnr1 expression in male and female PP groups, compared to their respective CC, CP and PC groups (*^/^** *p* < 0.05/0.01; ^#/##^
*p* < 0.05/0.01; ^&&&^
*p* < 0.001; Figure 3A). 

A sex effect was found in the mRNA levels of *Daglα* [F (1,34) = 7.883; *p* < 0.01] (Figure 3C).

Maternal and postnatal diets [F (1,41) = 4.099; *p*=0.051], and sex, maternal and postnatal diets interactions [F (1,41) = 6.522; *p* < 0.05], were found in the mRNA levels of endocannabinoid-synthesizing enzyme *Daglβ*, with the hypercaloric maternal diet increasing *Daglβ* mRNA levels in PC female offspring (* *p* < 0.05; ^&&^
*p* < 0.001; Figure 3D).

Three-way ANOVA also showed the effects of perinatal [F (1,41) = 12.057; *p* < 0.01] and postnatal diets [F (1,41) = 4.999; *p* < 0.05] on the mRNA levels of endocannabinoid synthesizing enzyme *Nape-pld*, with PP males and females having lower levels than respective CC, CP and PC groups (*^/^** *p* < 0.05/0.01; ^#^
*p* < 0.05; ^&&^
*p* < 0.01; Figure 3F).

### 3.4. Effects of Maternal and/or Postnatal Hypercaloric Diet on ECS and Glutamatergic Signaling Protein Levels in Adult Offspring Prefrontal Cortex

Regarding glutamate receptors, no main effects were found in the protein levels of GRIN1. However, Student´s *t* test showed an increase in GRIN1 in PFC of CP males compared to the control group (* *p* < 0.05; Figure 4A). The main effects of postnatal diet were found on the protein level of mGLUR5 [F (1,24) = 10.734; *p* < 0.01], with an increase in the mGLUR5 expression in the male CP group, compared to the control group, and a decrease in the PC males and females compared to the CP groups (* *p* < 0.05; ^#^
*p* < 0.05; Figure 4C). 

Regarding the endocannabinoid system, an effect of the maternal diet was found in the protein level of CB1 [F (1,24) = 7.597; *p* < 0.05], with the PC male group expressing significantly less CB1 that the control group (* *p* < 0.05; Figure 4D). An effect of the postnatal diet was found in the protein level of the cannabinoid receptor CB2 [F (1,24) = 9.489; *p* < 0.01], with an increase in the CB2 expression in the female CP group compared to the control group (* *p* < 0.05; Figure 4E). No main effects were found in the protein levels of DAGLβ. However, the Student´s t test showed an increase in DAGLβ in PFC of PP females, compared to the control group (* *p* < 0.05; Figure 4G). A significant interaction between maternal and postnatal diets and sex was found on the protein level of NAPE-PLD [F (1,24) = 4.877; *p* < 0.05], with an overall decrease in the NAPE-PLD protein level in CP and PC males (**^/^*** *p* < 0.01/0.001; Figure 4I). A significant interaction between maternal and postnatal diets was also found in the protein level of endocannabinoid-degrading enzyme FAAH [F (1,24) = 10.000; *p* < 0.01], with an overall decrease in the FAAH protein level in PC females (* *p* < 0.05; Figure 4J).

Appendix A shows membranes used in this study stained with Ponceau red stain for locating protein bands on Western blots, and the proteins analyzed in each one of them.

### 3.5. Effects of Maternal and/or Postnatal Hypercaloric Diet on Inflammation in Adult Offspring Prefrontal Cortex

Regarding gliosis, an interaction between sex and maternal diet was found in the mRNA levels of *Gfap* [F (1,41) = 4.605; *p* < 0.05] (Figure 5D).

No main effects were found in the mRNA expression of *Tlr4*. However, the Student´s *t* test showed an increase in *Tlr4* in PFC of the PC male, compared to the CP, group (^#^
*p* < 0.05; Figure 5A). 

### 3.6. Effects of Maternal and/or Postnatal Hypercaloric Diet on GABAergic Signaling in Adult Offspring Prefrontal Cortex

Regarding GABA receptors, only a sex effect was found on the mRNA levels of *Gabbr2* [F (1,41) = 4.558; *p* < 0.05], with an increase in *Gabbr2* mRNA levels in PP males compared to the CC group (* *p* < 0.05; Appendix A).

No more main effects were found, however the Student’s test showed an increase in the mRNA expression of *Gabrb2* and *Gabbr1* in PP males (** *p* < 0.01; ^#^
*p* < 0.05; Appendix A).

## 4. Discussion

The present study indicates that hypercaloric diet consumption during the perinatal period induces anxiety-related behavioral changes in adult offspring that kept consuming the palatable diet after weaning (PP animals). Furthermore, increased anxiety in PP animals is associated with specific changes in the prefrontal cortex’s (PFC) mRNA expression of ionotropic and metabotropic glutamate receptors, such as *Grin1*, *Gria1* and *Grm5*, and endocannabinoid signaling elements, such as the Cnr1 receptor or the endocannabinoid-synthesis/degrading enzymes Diacylglycerol lipase-beta (*Daglβ*) and N-acyl phosphatidylethanolamine phospholipase D (*Nape-pld*). The alterations suggest an enhanced glutamatergic activity in the PFC, associated with a reduced inhibitory input into the glutamate synapses through the cannabinoid CB1 receptor, whose expression was decreased. These changes are sex-dependent, being more marked in males.

Traditionally, maternal nutrition studies have focused on offspring metabolic diseases, such as glucose tolerance, insulin resistance, dyslipidaemia, hypertension and obesity, in humans and rodents [4,30,31]. However, it is known that continued stress, such as malnutrition with an excess or deficiency in most nutrients, impacting fetal neurodevelopment, can cause neuropsychiatric disorders, including depression, anxiety and attention-deficit/hyperactivity disorder in offspring, even though to adulthood [32]. Thus, both exposure to a high-fat diet and calorie restriction during the perinatal period can lead to eating disorders in offspring, such as hyperphagia and the preference for hypercaloric food [33,34,35]. In addition to eating disorders, perinatal exposure to a high-fat diet has been shown to increase anxiety-related behaviors in offspring, this effect being independent of the type of diet consumed during postnatal life [36,37,38].

Our results confirm that a free-choice palatable diet (P) during the perinatal period exerts an effect by itself on the offspring’s time in, and number of entries into, the central area of the open field test. However, we also observed an interaction between the maternal diet and the postnatal diets, affecting the same parameters mentioned, as well as the time in and number of entries into the open arms of the elevated plus maze test. Thus, the male PP group showed the most anxiety-like behaviors, associated with a decrease in these four behavioral indicators despite normal locomotion. These results also indicate a marked sexual dimorphism in long-term hypercaloric diet-induced anxiety behavior.Earlier studies reported the implications of a hypercaloric diet for brain structures and the neurotransmitter systems important in anxiety. For example, obese animals are susceptible to develop anxiety-like behaviors, probably through changes in the glutamatergic and GABAergic neurotransmission within the ventromedial and dorsomedial hypothalamic nucleus [39,40,41], which demonstrated that a high-fat/high-sugar diet induces alterations in the function of the prefrontal cortex through changes in the expression of GABAergic parvalbumin-expressing inhibitory interneurons, which was associated with increased anxiety-like behaviors.

Maternal malnutrition has also been shown to play an important role in programming behavioral disorders, such as anxiety, associated with alterations in the neurodevelopment of the brain areas involved in behavior. However, most studies are focused on the hypothalamic–pituitary–adrenal (HPA) axis [1,5,42]. The prefrontal cortex (PFC) provides executive control by coordinating cognitive, emotional and behavioral responses to threatening stimuli, however, the precise role of signaling within PFC, with respect to the stress of a maternal hypercaloric diet, has been minimally investigated.

Our results showed an overall activation of PFC glutamate signaling in the PP male group, with an increase in the mRNA levels of the glutamate synthesis enzyme *Gls*, the glutamate transporter *Slc1a1*, the ionotropic receptors NMDA type *Grin1* and *Grin2c*, the ionotropic receptors AMPA type *Gria1*, *Gria2* and *Gria4*, and the metabotropic receptors *Grm3* and *Grm5*. Several studies relate behavioral disorders in offspring, such as schizophrenia and anxiety-like behaviors, to alterations in GABAergic signaling (in involved areas such as HPA axis, hippocampus and cortex) induced by such perinatal insults as lipopolysaccharide (LPS), drugs and malnutrition [1,5,22,43]. Contrarily, our data show no effect of perinatal and/or postnatal free-choice P diet on GABAergic signaling in PFC of adult offspring. Furthermore, our results showed a negative correlation between anxiety-related behavioral tests and the main altered genes of the glutamatergic system. Thus, an increase in the expression of the glutamatergic system is associated with less time spent in the central area of the open field test, and less time in, and number of entries into, the open arms of the elevated plus maze test; that is, greater anxiety.

So, results from the present study suggest that the impact of the maternal and/or postnatal P diet on anxiety levels in adult offspring relate to alterations in PFC glutamate levels, or its modulatory systems, including the endocannabinoid system (ECS), as we will discuss below.

The ECS plays an important role in various physiological functions, including neuroprotection, synaptic plasticity and energy homeostasis. ECS is a neuromodulator of the major excitatory (glutamatergic) and inhibitory (GABAergic) neurotransmitter systems in the brain. In addition, the ECS plays a crucial role during critical periods of brain development, and its disruption by early stressful events, including metabolic disruptions, can lead to significant neuropsychiatric symptoms. [17]. Our results are in agreement with previous studies that demonstrate alterations in cannabinoid signaling in the brain of adult offspring induced by maternal malnutrition [8,24,28]. We found a decrease in the mRNA expression of the cannabinoid receptor *Cnr1* in the PFC of adult animals born from mothers fed with a free-choice palatable diet, and who continued eating the P diet after weaning. Furthermore, the protein levels of CB1 are also affected by the maternal diet, with an overall decrease in its expression in males compared to the control group. The palatable diet consumed from the perinatal period to adulthood (PP groups) also induces a decrease in gene levels of *Nape-pld*, the main synthesis enzyme of anandamide (AEA). The decrease of both *Cnr1* and *Nape-pld* in the PP groups could indicate a decrease in AEA, that would explain the increase in anxiety in these animals, since the anxiolytic role of this endocannabinoid has been demonstrated, and the pharmacological augmentation of central endogenous cannabinoid (eCB) signaling being a promising treatment for anxiety disorders [44].

Our results also discard a potential diet-associated neuroinflammation factor in these observed long-term alterations, despite previous reports suggesting that a hypercaloric diet might result in neuroinflammation, thus contributing to behavioral disorders [40,45]. Our results do not support this hypothesis, since we did not find alterations in the mRNA expression of several genes that would indicate the activation of inflammatory pathways, such as the toll-like receptor related to pathogen recognition and the activation of innate immunity *Tlr4*, the cyclooxygenase key in prostaglandin biosynthesis *Ptgs2*, the microglia/macrophage-specific calcium-binding protein *Iba1*, the major intermediate filament proteins of mature astrocytes *Gfap*, and the cannabinoid receptor *Cnr2*, induced by the perinatal and/or postnatal P diet in the adult offspring PFC.

The perinatal and/or postnatal hypercaloric diet can exert a direct effect on cannabinoid signaling, decreasing cannabinoid tone, interrupting the negative feedback that this system exerts on glutamatergic neurotransmission, which is thus increased. There could also be a direct effect of diet on glutamatergic signaling, by increasing it. Both effects could explain the increase in the anxious behavior of PP animals. Although we have not addressed the impact of specific nutrients as potential sources for the effects observed, we must consider that the fatty acid composition of the diet might dysregulate the amount of endocannabinoid precursors [46], and that maternal hypoproteic diets can also disrupt brain glutamatergic systems [47].

In the present study we have found some inconsistencies between the genetic and protein results. We consider these differences to be a limitation of the study, due to the low number of samples analyzed by western blot. So, future extensive protein studies will be necessary to elucidate this controversy.

## 5. Conclusions

The conditions of early life contribute greatly to the risk of mental disorders, and therefore, the study of the origins and mechanisms involved in the emergence of these disorders is essential for their prevention and/or treatment.

In this study, we demonstrate that a free-choice palatable diet during the perinatal period induces changes in the glutamatergic signaling of the adult offspring prefrontal cortex, associated with alterations in the endocannabinoid system, a main regulator of the glutamate synapses. Furthermore, these changes are more evident when the palatable diet is maintained after weaning, and its effects are sex-dependent.

Glutamatergic and endocannabinoid disorders induced by maternal diet are related to increased anxiety-like behavior in adulthood.

## Figures and Tables

**Figure 1 nutrients-12-01829-f001:**
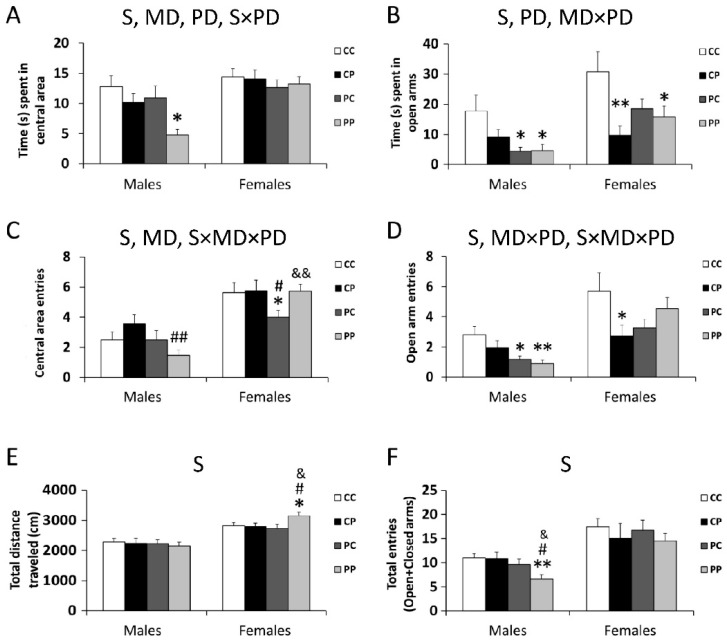
Effect of maternal and/or offspring exposure to a free-choice palatable (P) diet on anxiety-related behavior in the elevated plus maze test and in the open field test: time spent in central area of the open field test (**A**), time spent in open arms of the elevated plus maze test (**B**), number of entries into central area of the open field test (**C**), number of entries into the open arms of the elevated plus maze test (**D**), total distance traveled in the open field test (**E**), total entries into the open and closed arms of elevated plus maze test (**F**). Data are expressed as the mean ± S.E.M. (*n* = 6). Student’s *t* test: *^/^** *p* < 0.05/0.01 vs. CC males or CC females; ^#/##^
*p* < 0.05/0.01 vs. CP males or CP females; ^&/&&^
*p* < 0.05/0.01 vs. PC males or PC females. MD: maternal diet effect; PD: postnatal diet effect; SxPD: sex and postnatal diet interaction; MDxPD: maternal and postnatal diets interaction; SxMDxPD: sex, maternal diet and postnatal diet interaction.

**Figure 2 nutrients-12-01829-f002:**
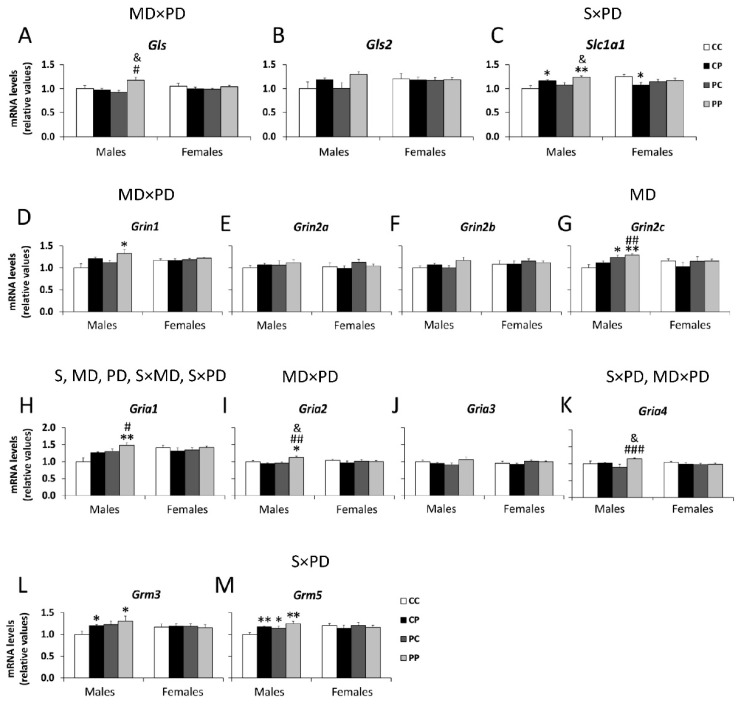
Effect of maternal and/or offspring exposure to a free-choice palatable (P) diet on the relative mRNA levels of glutamatergic signaling genes: *Gls* (**A**), *Gls2* (**B**), *Slc1a1* (**C**), *Grin1* (**D**), *Grin2a* (**E**), *Grin2b* (**F**), *Gin2c* (**G**), *Gria1* (**H**), *Gria2* (**I**), *Gria3* (**J**), *Gria4* (**K**), *Grm3* (**L**) and *Grm5* (**M**) in the prefrontal cortex of male and female offspring in adulthood. Data are expressed as the mean ± S.E.M. (*n* = 6). Student’s *t* test: *^/^** *p* < 0.05/0.01 vs. CC males or CC females; ^#/##/###^
*p* < 0.05/0.01/0.001 vs. CP males or CP females; ^&^
*p* < 0.05 vs. PC males or PC females. MD: maternal diet effect; PD: postnatal diet effect; S: sex effect; SxMD: sex and maternal diet interaction; SxPD: sex and postnatal diet interaction; MDxPD: maternal and postnatal diets interaction.

**Figure 3 nutrients-12-01829-f003:**
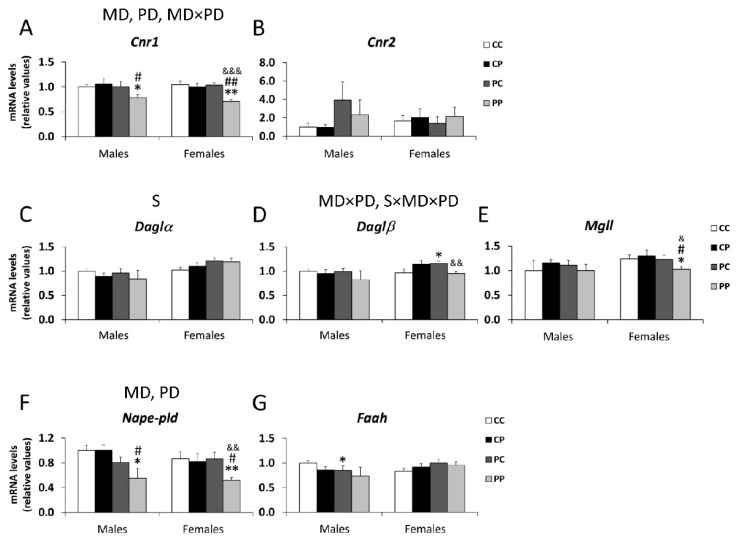
Effect of maternal and/or offspring exposure to a free-choice palatable (P) diet on the relative mRNA levels of endocannabinoid system genes: *Cnr1* (**A**), *Cnr2* (**B**), *Dagl**α* (**C**), *Dagl**β* (**D**), *Mgll* (**E**), *Nape-pld* (**F**) and *Faah* (**G**) in the prefrontal cortex of male and female offspring in adulthood. Data are expressed as the mean ± S.E.M. (*n* = 6). Student´s t test: *^/^** *p* < 0.05/0.01 vs. CC males or CC females; ^#/##^
*p* < 0.05/0.01 vs. CP males or CP females; ^&/&&/&&&^
*p* < 0.05/0.01/0.001 vs. PC males or PC females. MD: maternal diet effect; PD: postnatal diet effect; S: sex effect; MDxPD: maternal and postnatal diets interaction; SxMDxPD: sex, maternal diet and postnatal diet interaction.

**Figure 4 nutrients-12-01829-f004:**
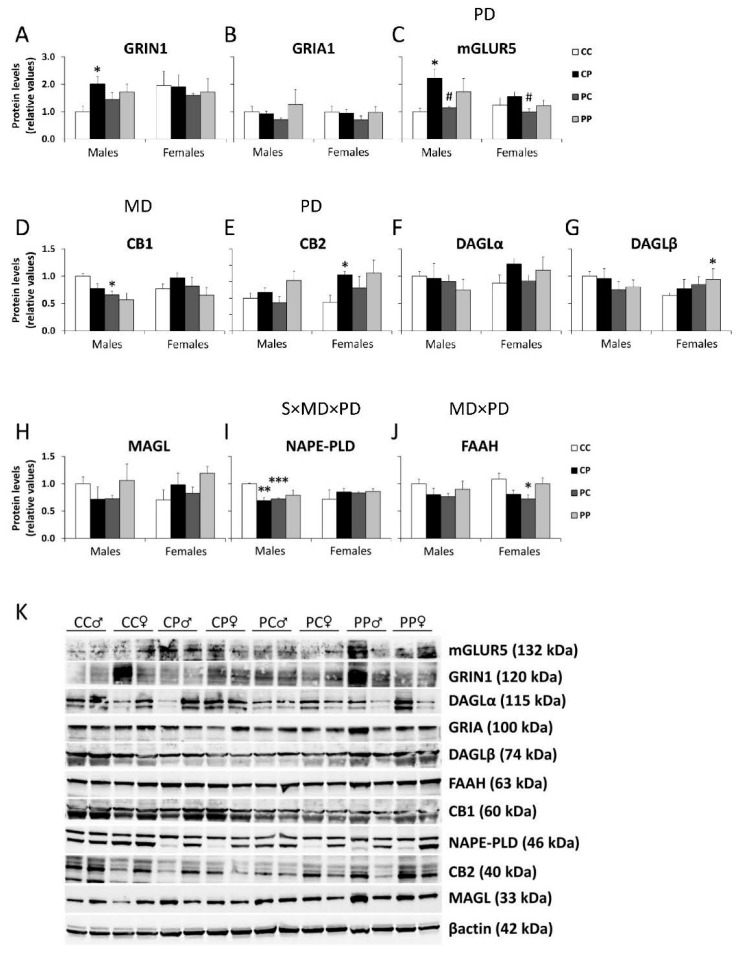
Effect of maternal and/or offspring exposure to a free-choice palatable (P) diet on the relative protein levels of: (**A**), GRIA1 (**B**), mGLUR5 (**C**), CB1 (**D**), CB2 (**E**), DAGLα (**F**), DAGLβ (**G**), MAGL (**H**), NAPE-PLD (**I**) and FAAH (**J**) in the prefrontal cortex of male and female offspring in adulthood. Representative immunoblots (**K**). Data are expressed as the mean ± S.E.M. (*n* = 4). Student’s *t* test: *^/^**^/^*** *p* < 0.05/0.01/0.001 vs. CC males or CC females; ^#^
*p* < 0.05 vs. CP males or CP females. MD: maternal diet effect; PD: postnatal diet effect; MDxPD: maternal and postnatal diets interaction; SxMDxPD: sex, maternal diet and postnatal diet interaction.

**Figure 5 nutrients-12-01829-f005:**
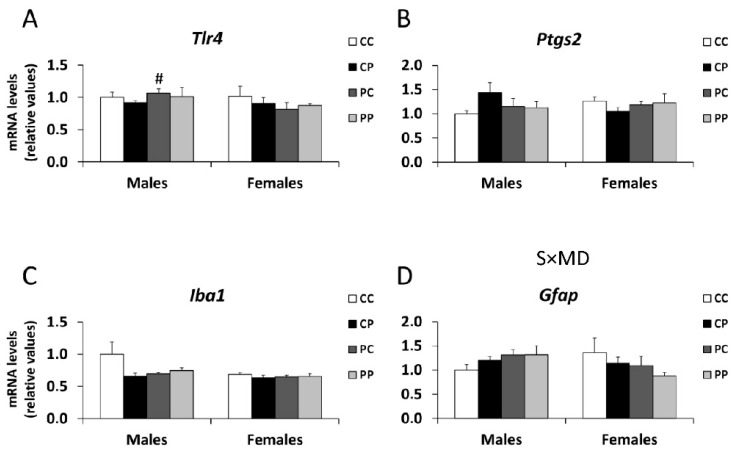
Effect of maternal and/or offspring exposure to a free-choice palatable (P) diet on the relative mRNA levels of: *Tlr4* (**A**), *Ptgs2* (**B**), *Iba1* (**C**) and *Gfap* (**D**) in the prefrontal cortex of male and female offspring in adulthood. Data are expressed as the mean ± S.E.M. (*n* = 6). Student´s *t* test: ^#^
*p* < 0.05 vs. CP males or CP females. SxMD: sex and maternal diet interaction.

**Table 1 nutrients-12-01829-t001:** Diet composition.

	Protein	Total Carbohydrate	Simple Carbohydrates	Total Fat	Saturated Fatty Acid	Unsaturated Fatty Acid	Fiber	Sodium	Energy
Standard chow diet	16.1%	60.0%	3.3%	3.1%	22.20%	77.70%	4.0%	0.003%	2.9 Kcal/g
Palatable diet	6.0%	60.4%	89.0%	24.45%	56.2%	43.88%	1.45%	0.17%	4.88 Kcal/g

**Table 2 nutrients-12-01829-t002:** Primer references for Taqman^®^ Gene Expression Assays.

Gene Symbol	Assay ID	GenBank Accession Number	Amplicon Length (bp)
*Actb*	Rn00667869_m1	NM_0311443	91
*Cnr1 (CB1)*	Rn02758689_s1	NM_012784.4	92
*Cnr2 (CB2)*	Rn01637601_m1	NM_020543.4	68
*Napepld*	Rn01786262_m1	NM_199381.1	71
*Faah*	Rn00577086_m1	NM_024132.3	63
*Daglα*	Rn01454304_m1	NM_001005886.1	67
*Daglβ*	Rn01453771_m1	NM_001107120.1	98
*Mgll (MAGL)*	Rn00593297_m1	NM_138502.2	78
*Gls (kidney-type)*	Rn00561285_m1	NM_001109968.1	80
*Gls2 (liver-type)*	Rn00594296_m1	NM_001270786.1	58
*Grin1 (GluN1)*	Rn01436034_m1	NM_001270602.1	73
*Grin2a*	Rn00561341_m1	NM_012573.3	68
*Grin2b*	Rn00680474_m1	NM_012574.1	79
*Grin2c*	Rn00561359_m1	NM_012575.3	57
*Gria1 (GluA1)*	Rn00709588_m1	NM_031608.1	85
*Gria2*	Rn00568514_m1	NM_001083811.1	122
*Gria3*	Rn00583547_m1	NM_001112742.1	74
*Gria4*	Rn00568544_m1	NM_001113184.1	76
*Grm3*	Rn01755349_m1	NM_001105712.1	71
*Grm5 (mGlu5)*	Rn005666628_m1	NM_017012.1	112
*Slc1a1 (EAAAC1)*	Rn00564705_m1	NM_013032.3	92
*Gabra1*	Rn00788315_m1	NM_183326.2	75
*Gabra2*	Rn01413643_m1	NM_001135779.1	123
*Gabrb1*	Rn00564146_m1	NM_012956.1	81
*Gabrb2*	Rn00564149_m1	NM_012957.2	64
*Gabrg1*	Rn00589841_m1	NM_080586.1	73
*Gabrg2*	Rn01464079_m1	NM_183327.1	78
*Gabbr1*	Rn00578911_m1	NM_031028.3	113
*Gabbr2*	Rn00582550_m1	NM_031802.1	87
*Tlr4*	Rn00569848_m1	NM_019178.1	127
*Ptgs2*	Rn01483828_m1	NM_017232.3	112
*Iba1*	Rn00574125_g1	NM_017196.3	126
*Gfap*	Rn01253033_m1	NM_017009.2	75

**Table 3 nutrients-12-01829-t003:** Antibodies used for protein expression by Western blotting.

Antigen	Immunogen	Manufacturing Details	Dilution
**βactin**	Slightly modified β-cytoplasmic actin N-terminal peptide, Ac-Asp-Asp-Asp-Ile-Ala-Ala-Leu-Val-Ile-Asp-Asn-Gly-Ser-Gly-Lys, conjugated to KLH.	Sigma #2535L, Mouse monoclonal antibody.	1:2000
**CB1**	Synthetic peptide corresponding to C terminal amino acids 461-472 of human cannabinoid receptor I (MSVSTDTSAEAL)	Abcam. Rabbit polyclonal antibody. Cat. nº: ab23703	1:200
**CB2**	Fusion protein, corresponding to aa 1-32 of rat cannabinoid receptor II	Abcam. Rabbit monoclonal antibody. Cat. nº: ab3561	1:200
**DAGLα**	KLH conjugated synthetic peptide derived from the region aa121-195 of human diacylglycerol lipase alpha (DAGLA)	bioNova Científica (#orb156533). Rabbit polyclonal antibody	1:100
**DAGLβ**	KLH conjugated synthetic peptide derived between 61-140 amino acids of human diacylglycerol lipase beta (DAGLB)	Biorbyt (orb182976). Rabbit polyclonal antibody	1:200
**MAGL**	Recombinant fragment corresponding to Human Monoacylglycerol Lipase/MGL aa 1-14.	Abcam. Rabbit polyclonal antibody. (Ab24701)	1:200
**FAAH**	Synthetic peptide from rat Fatty acid amide hydrolase (FAAH), aa 561-579 (CLRFMREVEQLMTPQKQPS)	Cayman. Rabbit polyclonal antibody. Cat. Nº: 101600	1:200
**NAPE-PLD**	Synthetic peptide:YMGPKRFRRSPCTI, corresponding to amino acids 159–172 of Human N-acyl phosphatidylethanolamine phospholipase D (NAPE-PLD) (UniProt: Q6IQ20)	Abcam. Rabbit polyclonal antibody. Ab95397	1:200
**NMDAR1**	Synthetic peptide corresponding to the C-terminus of rat N-methyl-D-aspartate (NMDA) receptor subunit (Catalog number AG344).	Sigma. Rabbit monoclonal antibody. AB9864	1:200
**GluR1**	Synthetic phospho-peptide corresponding to amino acid residues surrounding Ser845 of rat Glutamate Ionotropic Receptor AMPA Type Subunit 1 (GRIA1) conjugated to KLH	Thermo Fisher. Rabbit polyclonal antibody. OPA1-04118	1:200
**mGluR5**	Synthetic peptide corresponding to residues Y D R R L A Q H K S E I E of Metabotropic glutamate receptor 5 (mGluR5).	Thermo Fisher. Rabbit polyclonal antibody. (#PA1-24637)	1:500

**Table 4 nutrients-12-01829-t004:** Pearson’s correlation coefficients between behavioral parameters related to anxiety and elements of the glutamatergic system.

	Gls	Slc1a1	Grin1	Gria1	Gria2	Grm3	Grm5
**Time spent** **in central area**	−0.622 **				−0.653 **		
**Time spent** **in open arms**				−0.424 *			−0.517 *
**Open arm entries**		−0.513 *	−0.541 **	−0.462 *		−0.443 *	−0.598 **

* *p* < 0.05; ** *p* < 0.01.

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
