# Peer review of "Sex-Specific Anxiety and Prefrontal Cortex Glutamatergic Dysregulation Are Long-Term Consequences of Pre-and Postnatal Exposure to Hypercaloric Diet in a Rat Model"

_nutrients, 2020, doi:10.3390/nu12061829_

Round 1

Reviewer 1 Report

The manuscript "Sex-specific anxiety and prefrontal cortex 2 glutamatergic dysregulation are long term 3 consequences of pre-and postnatal exposure to 4 hypercaloric diet" presents  quite novel and interesting results on the sex-dependent effect of hypercaloric diet on glutamatergic and GABAergic transmission as well as endocannabinoid signalling in the prefrontal cortex.

The paper is well written  and will certainly contribute to the scientific community. There is just a remark regardint the graphs: their number is huge. To simplify the reading, the three way ANOVA main effects and/or interaction should be indicated within each graphs using the factors' initial/s (S, MD, O, SxMD and so on...).

Reviewer 2 Report

This is a well performed study designed to assess anxiety behaviours in combination with gene and protein expression changes in glutamatergic, GABAergic, and endocannabinoid signalling in the rat prefrontal cortex in response to hypercaloric diet. A comprehensive, well controlled approach is used to assess the impact of both maternal and offspring diets. I commend the authors on the clarity of their writing, good sample sizes across all experiments, and detailed methods. The study will be useful to the field. However, I have a few concerns that the authors should address:

  • This paper contains a lot of data linked to individual animals, and it would therefore seriously benefit from a more integrated statistical approach to better understand and visualise key patterns that do not necessarily emerge when simply looking at all the figures separately. To improve this, I suggest the authors consider principal component analysis of their data. Perhaps even correlative analyses of key genes/proteins with the anxiety behaviours would prove useful.
  • Several claims are not supported by the data and need amending – in particular, relating to the western blot analysis of protein levels. No significant differences between control (CC) and PP-treated mice are observed for any protein in male mice, and only DAGLBeta in female mice. Hence, a sentence in the abstract (Lines 20-23) and discussion (Lines 353-357, Line 410) need amending. Similarly, Dag1 levels are not changed compared to control (Line 356), there are in fact some differences in GABAergic signalling genes (Lines 398-399), and CB1 and NAPE protein levels are not changed in PP males (Line 23-25).
  • The model used, i.e. rat, needs to be included in both the title and abstract.
  • Why were t-tests and not post-hoc tests used? Were the multiple t-tests Bonferroni corrected? If not, they should be to reduce the chances of Type I error.
  • There appears to be a stark lack of congruence between RNA and protein levels – please comment on this, and highlight as a possible limitation to the work.

Round 2

Reviewer 2 Report

The authors have done an excellent job at responding to my suggestions. I like the additions and changes, and I hope the authors think that the edits have improved the manuscript. Well done.